# Virological characterization of a new isolated strain of Andes virus involved in the recent person-to-person transmission outbreak reported in Argentina

Rocio Coelho[1☯], Sebastian Kehl[1☯]*, Natalia Periolo[1], Emiliano Biondo[2], Daniel Alonso[1], Celeste Perez[1], Darío Fernández Do Porto[3], Gustavo Palacios[4], Alexis Edelstein[1], Carla Bellomo[1], Valeria Paula Martinez[1]

**1** Administración Nacional de Laboratorios e Institutos de Salud "Dr. C. G. Malbrán", Buenos Aires, Argentina, **2** Área programática de Esquel, Secretaria de Salud Chubut, Esquel, Argentina, **3** Universidad de Buenos Aires, Facultad de Ciencias Exactas y Naturales (FCEyN-UBA), Universidad de Buenos Aires, CONICET, Buenos Aires, Argentina, **4** Icahn School of Medicine at Mount Sinai, New York, United States of America

☯ These authors contributed equally to this work.

* skehl@anlis.gob.ar

## Abstract

On November 2, 2018, a person-to-person transmission outbreak of Andes virus (*Orthohantavirus andesense)* began in the small town of Epuyén, Argentina. The strain demonstrated a high capacity for sustained transmission among the human population requiring the implementation of quarantine measures, rigorous contact tracing, isolation of close contacts, and active clinical monitoring to prevent further spread. In this study, we report the isolation of this strain, which we name the ARG-Epuyén strain, directly from a clinical sample after just three passages in cell culture. Complete sequencing revealed only a single amino acid change post-isolation, suggesting that this strain can be considered a non-adapted wild-type Andes virus, marking a critical step toward the development of medical countermeasures against this emerging pathogen. The pathogenicity and transmissibility potential of ARG-Epuyén were evaluated in hamsters, the only animal model for Hantavirus Pulmonary Syndrome. Additionally, this strain was compared with Andes/ARG, an ANDV strain previously isolated from the same geographical area in the Argentinian Patagonia, from a rodent specimen. Our findings revealed high infectiousness and efficient hamster-to-hamster transmission through direct contact experiments, although ARG-Epuyén appeared to be less pathogenic than Andes/ARG.

## Author summary

Hantavirus Pulmonary Syndrome (HPS) in Argentina is a devastating disease with a fatality rates up to 50%. Andes virus (ANDV) is the most frequent

**Data availability statement:** All relevant data are within the manuscript and its Supporting Information files.

**Funding:** The author(s) received no specific funding for this work.

**Competing interests:** The authors have declared that no competing interests exist.

causative agent of this infection in Argentina, and was associated with two outbreaks of sustained person-to-person transmission. The paucity of studies for vaccine development and other medical countermeasures (MCMs) against hantaviruses was partially due to the limited access to wild type viral strains and the lack of animal models of diseases. American hantaviruses are very difficult to isolate "*in vitro*" and "*in vivo*", and there are only three strains of ANDV from Chile and Argentina that could be propagated in cell culture, but none of them were associated to person-to-person transmission. Also, the ANDV strain most used in "*in vitro*" and "*in vivo*" studies were not involved in human disease and has a unclear history of cell passages. In our work we report the isolation of an ANDV strain associated with disease in humans but, most importantly, with person-to-person transmission (Epuyén outbreak in 2018). In this article we also describe the results of the characterization of this strain proving its ability to infect and spread efficiently between hamsters. Our work represents a critical step towards the development of MCMs against this emerging pathogen.

## Introduction

Hantaviruses (*Bunyaviricetes: Elliovirales: Hantaviridae: Mammantavirinae*) are enveloped, single stranded, negative sense RNA viruses with three-segmented genome. The genomic segments consist of a small segment (S), a medium segment (M), and a large segment (L), which encode the nucleocapsid (N) protein, a nonstructural protein (NSs) in some species, surface glycoproteins (Gn and Gc), and an RNA-dependent RNA polymerase (RdRp), respectively [1]. Hantaviruses are distributed worldwide and are hosted by various vertebrate animal species. Pathogenic hantaviruses are primarily associated with rodents as natural reservoirs and are classified under the genus *Orthohantavirus*. These viruses establish seemingly asymptomatic and chronic infections in several rodent species. The risks of viral spillover have increased due to new farming practices, climate change, the expansion of rural human settlements, and disruptions to the zoonotic interface. Additionally, rural tourism has led to travel-related cases [2–4].

Several species of orthohantaviruses are responsible for Hantavirus Pulmonary Syndrome (HPS) in the Americas and Hemorrhagic Fever with Renal Syndrome (HFRS) in Asia and Europe. HPS, first described in 1993 in the US [5], is caused by at least 24 distinct viruses [6]. In Argentina, most HPS cases are caused by 7 viruses closely related to Andes virus (ANDV), species *Orthohantavirus andesense*. ANDV was the first hantavirus characterised in Argentina [7]. It was associated with the long-tailed pygmy rice rat *Oligoryzomys longicaudatus* in the Patagonian Andean region. After human infection, the signs and symptoms of the disease can manifest after a long period of up to 40 days [8,9]. Severe cases had progressive pulmonary edema, hypoxia and hypotension; fatal cases had a severe compromise in hemodynamic function. ANDV-HPS is associated with high case-lethality rates ranging from 21–50% [10,11].

Humans generally become infected through the inhalation of aerosolized rodent excreta. Before 1996, the route of orthohantavirus transmission was considered strictly zoonotic, resulting in "dead-end" human infections [7]. However, in 1996, an ANDV-caused HPS outbreak occurred in the small city of El Bolsón and then expanded to distant cities, such as Bariloche (121 km) and Buenos Aires (1700 km), involving 16 epidemiologically linked cases. This outbreak became a focal point for orthohantavirus research because molecular and epidemiological evidence suggested person-to-person (PTP) transmission [12,13]. A larger PTP transmission outbreak that began in 2018 and involved 34 cases and was curtailed by the implementation of strict quarantine measures. In this outbreak, several individuals were identified as superspreaders, predicting the high transmission potential of this strain [10].

New World Hantaviruses are very difficult to isolate in cell culture. Only three strains of ANDV had been propagated in cell culture (CH-9717869; CH-7913 and Andes/ARG). CHI-7913 was obtained from a serum sample of a Chilean patient before the onset of symptoms [14]. CH-9717869 and Andes/ARG were isolated from long tailed pygmy rice rats captured in Chile and Argentina respectively [15,16]. In 2001, it was first described that CH-9717869 caused lethality and a disease very similar to HPS in Syrian Golden Hamsters, an unusual finding for orthohantaviruses at the time [17], and the model ANDV/hamster became a unique resource for the study of HPS. We were later able to reproduce this highly lethal model with the Argentinean rodent strain Andes/ARG [18]. This was intriguing considering that reservoir infection with hantavirus is considered mostly asymptomatic. Moreover, the human CHI-7913 strain resulted in an asymptomatic infection in the same animal model suggesting that the lethal ANDV model of HPS is strain specific, reinforcing the necessity to obtain and characterise new human strains of ANDV to understand their pathogenesis in humans [19]. In this work, we report the isolation in cell culture of the strain responsible for the largest ANDV PTP-transmission outbreak ever reported, ARG-Epuyén. In addition, we performed a preliminary virological characterization of this strain proving its ability to infect hamsters causing high levels of viremia and to spread efficiently between them through direct contact experiments.

## Materials and methods

### Biosafety precautions and Ethics statement

The entire procedure of viral isolation, subsequent propagation, and sample analysis before viral inactivation was conducted at the biosafety level 3 facility at the Unidad Operativa Centro de Contención Biológica (UOCCB-ANLIS). All procedures involving animal handling were carried out within an animal biosafety level 3 facility at UOCCB-ANLIS. The hamsters were housed in ventilated cage racks. To ensure compliance with ethical standards [20], procedures were approved by Comité de Bioética Comodoro Rivadavia, Secretaría de Salud, Chubut province, Argentina, under the protocol number 4/2024.

### Case description and sample selection

During a PTP outbreak from November 2018 to March 2019, residents of Epuyén (Chubut province, Argentina) and neighbouring areas were invited by the Ministry of Health of Chubut province to participate in a seroprevalence study to assess the previous circulation of the virus. Written informed consent was obtained from all the volunteers. One of them, who had recently been exposed to a previous case of HPS, developed symptoms on December 28, 2018, and was subsequently hospitalized. Hantavirus infection was confirmed through the detection of IgM and IgG antibodies specific to ANDV by ELISA [21], as well as the presence of viral RNA via RT-qPCR. The sample collected for the seroprevalence study one day before symptom onset (day -1) was later analyzed to quantify viral load and antibody titres. Complete genome sequencing revealed that this patient was among the 33 cases involved in the PTP transmission outbreak [8].

### Virus isolation on cell culture

The serum sample collected on day -1 was filtered in a membrane of 0.2 (Millipore) and diluted in a concentration of 1:10. Then was inoculated onto a Vero E6 cell monolayer (CRL-1586; ATCC, Manassas, VA, USA) and incubated for one hour

at 37°C in a humidified atmosphere containing 5% CO2. After the incubation, fresh complete medium (MEM, 10% FBS, antibiotics, and antimycotics) was added to the T25 flask, which was then incubated at 37°C under the same conditions. After 15 days, the cells were trypsinized, washed, and seeded onto a new monolayer, which was incubated under identical conditions (first blind passage). One-third of the cells were stored at -80°C, another third was tested by indirect immuno-fluorescence assay (IFA), and the final third was seeded onto a new monolayer. The medium was centrifuged, aliquoted, and stored at -80°C. Three blind passages were performed in total. Each passage was monitored by real-time RT-PCR for ANDV RNA in the culture media and by IFA for detecting viral antigen in the cells. Mock-infected Vero E6 cells, treated under identical conditions, were used as a negative control. After confirming the isolation, a fresh Vero E6 cell monolayer was infected to produce a large viral stock (fourth passage, p4). The supernatant was collected daily over a 10-day period, centrifuged, and titrated by focus-forming assay (FFA) as previously described [9]. Supernatants with the highest titers were pooled, aliquoted, and stored at -80°C. This newly isolated strain was subsequently designated as ARG-Epuyén.

### Immunofluorescence assay

Infected and mock-infected cells were resuspended in PBS and washed three times by centrifugation at 400 x g for five minutes at 4°C. The pellets were then resuspended in 1 ml PBS, and several 10 µl drops of each suspension were spotted onto a slide. The slides were left inside a laminar flow cabinet until completely dried. The slides were then immersed in ice-cold acetone for 10 minutes at -20°C, removed from the acetone, allowed to dry, and stored at -20°C until further processing. The slides were blocked with 3% horse serum in PBS-Triton for 15 minutes at room temperature and then washed with PBS. The cells were treated with a 1:500 dilution of rabbit polyclonal serum against ANDV in PBS-Triton-BSA and incubated for one hour at 37°C in a humid chamber [21]. After washing, the slides were incubated with FITC-conjugated anti-rabbit IgG (Kirkegaard & Perry) in PBS-Triton-BSA under the same conditions. Finally, the cells were incubated with 1 µg/ml DAPI in PBS for 15 minutes at room temperature and then washed. The slides were allowed to drain, mounted, and analyzed under a fluorescent microscope.

### Virus infection in Vero E6 and A549 cell lines

Vero E6 (CRL-1586; ATCC, Manassas, VA, USA) and A549 (human epithelial lung cell, ATCC CCL-185) cells were maintained in Dulbecco's Modified Eagle's Medium (DMEM high glucose; Sigma) supplemented with 5% fetal bovine serum (FBS, Gibco), 10 mM HEPES buffer, and 2 mM L-glutamine. The cells were infected with Andes/ARG and ARG-Epuyén at the indicated multiplicity of infection (MOI) for 60 minutes at 37°C. After incubation, the monolayers were washed three times with phosphate-buffered saline (PBS), and the cells were maintained in complete DMEM (Sigma). Supernatants were collected for 6 days after infection, centrifuged to remove cells, and stored at -80°C until use. Viral RNA was isolated using the Qiagen QIAamp Viral RNA kit according to the manufacturer's protocol. Viral growth kinetics were determined by quantifying ANDV RNA at different days post-infection using real-time RT-PCR as described below.

### Evaluation of ARG-Epuyén virulence in the hamster model

To evaluate whether ANDV could be transmitted between hamsters in a manner similar to human transmission, Syrian Golden Hamsters (*Mesocricetus auratus)* were infected with either the ARG-Epuyén or Andes/ARG strains. Eighteen 9-week-old hamsters were distributed across seven ventilated cages. One hamster per cage was infected via intramuscular injection (rear thigh) with 100 µl ($10^4$ FFU) of ARG-Epuyén passage No. 4 (two males and four females) or Andes/ARG passage No. 19 (one male). The infected hamsters were placed in direct contact with one or two non-infected individuals in the same ventilated cages on the day they were inoculated (day 0). As a control group, four hamsters were inoculated with cell culture supernatant from non-infected Vero E6 cells. All individuals were monitored daily for 30 days for signs of disease, including fatigue, inappetence, lethargy (reluctance to move), and/or dyspnea. Animals with fatal outcomes were necropsied on the day of death to collect lung samples, which were stored at -80°C until processing. On day 30, all

survivors were anesthetized by inhalation of isoflurane, terminally bled by cardiac puncture, and necropsied to obtain lung tissue.

### Enzyme linked immunosorbent assay

The detection of IgG antibodies against the viral nucleoprotein (NP) in hamsters was performed by ELISA, as previously described, to confirm infection [18]. Briefly, serial dilutions of hamster sera were incubated in polystyrene plates coated with recombinant ANDV-NP and a nonspecific recombinant protein in a humid chamber at 37°C for 1 hour. After washing, peroxidase-labelled goat anti-hamster IgG (H+L) (Kirkegaard & Perry) was added and incubated under the same conditions. TMB solution was used as a substrate, and optical density (OD) was measured at 450 nm. ΔOD values were calculated by subtracting the OD measured for each sample incubated with the specific (ANDV-NP) and nonspecific recombinant proteins. Samples were considered IgG positive if ΔOD values were greater than 0.4. The IgG titre was calculated as the inverse of the highest dilution that yielded a positive result.

### Viral RNA detection and genomic sequencing

Total RNA was extracted from lung samples and from the culture medium of infected cell passages using TRIzol (Invitrogen) according to the manufacturer's protocol. RT-qPCR was performed as previously described [22]. Briefly, each RNA sample was amplified in duplicate using the One Step RT-qPCR qScript kit (Quanta Biosciences) following the manufacturer's instructions. Each reaction mixture contained 12.5 µl of 2X Master Mix, 2 µl of an oligonucleotide mix designed to amplify the ANDV S-segment (1 µM each), 4.75 µl of nuclease-free water, 0.5 µl of the ANDV probe (5´FAM-BHQ-3´), 0.25 µl of qScript One Step RT, and 5 µl of template RNA. Reverse transcription was performed at 50°C for 15 minutes, followed by denaturation at 95°C for 5 minutes, and 40 cycles of amplification at 95°C for 15 seconds and 60°C for 60 seconds in a Real-Time PCR Detection System CFX-96 (Bio-Rad, CA). To obtain complete genomic sequences, an amplicon-based method was selected using a one-step RT-PCR strategy to enrich vRNA, followed by library preparation as previously described [23]. The library was sequenced on a MiSeq sequencing platform (Illumina, San Diego, CA) using 2 x 151-bp paired-end sequencing. Bioinformatic analysis was performed as described previously [8].

## Results

### A new strain of Andes virus was isolated in cell culture from a human serum sample

A serum sample obtained from an HPS case during the incubation period, with a high viral load ($6.3 \times 10^7$ copies/ml) and no detectable specific antibodies against ANDV, was used to infect a Vero E6 cell monolayer. After three blind passages, viral RNA was detectable in the supernatant of the infected cell culture ($2.5 \times 10^8$ copies/ml). Infection was confirmed in the cells by IFA, showing the presence of viral antigen in approximately 50% of the cells in passage No. 3 and 80% in passage No. 4 (p4) (Fig 1). A large viral stock was prepared from the supernatant of passage No. 3, as described above, for use in subsequent experiments. The infectious titre of the viral stock obtained was $2.1 \times 10^6$ FFU/ml. This new strain was designated as ARG-Epuyén.

### ARG-Epuyén and Andes/ARG showed differential kinetics of replication and infectivity in cell cultures

The growth kinetics of the Andes/ARG and ARG-Epuyén strains were first assessed in Vero E6 cells using an MOI of 0.002 (Fig 2A). Quantification of viral RNA in the culture supernatants of cells infected with each strain showed that Andes/ARG replicates faster than ARG-Epuyén. Both strains produced high levels of infectious viral particles, with the maximum level observed for Andes/ARG, which reached up to $2.1 \times 10^7$ FFU/ml, whereas ARG-Epuyén did not exceed $2.1 \times 10^6$ FFU/ml (Fig 2B). The growth kinetics were then compared using the epithelial cell line A549 with MOIs of 1. The results were consistent, showing that Andes/ARG replicates faster than ARG-Epuyén in both Vero E6 and A549 cells (Fig 3).

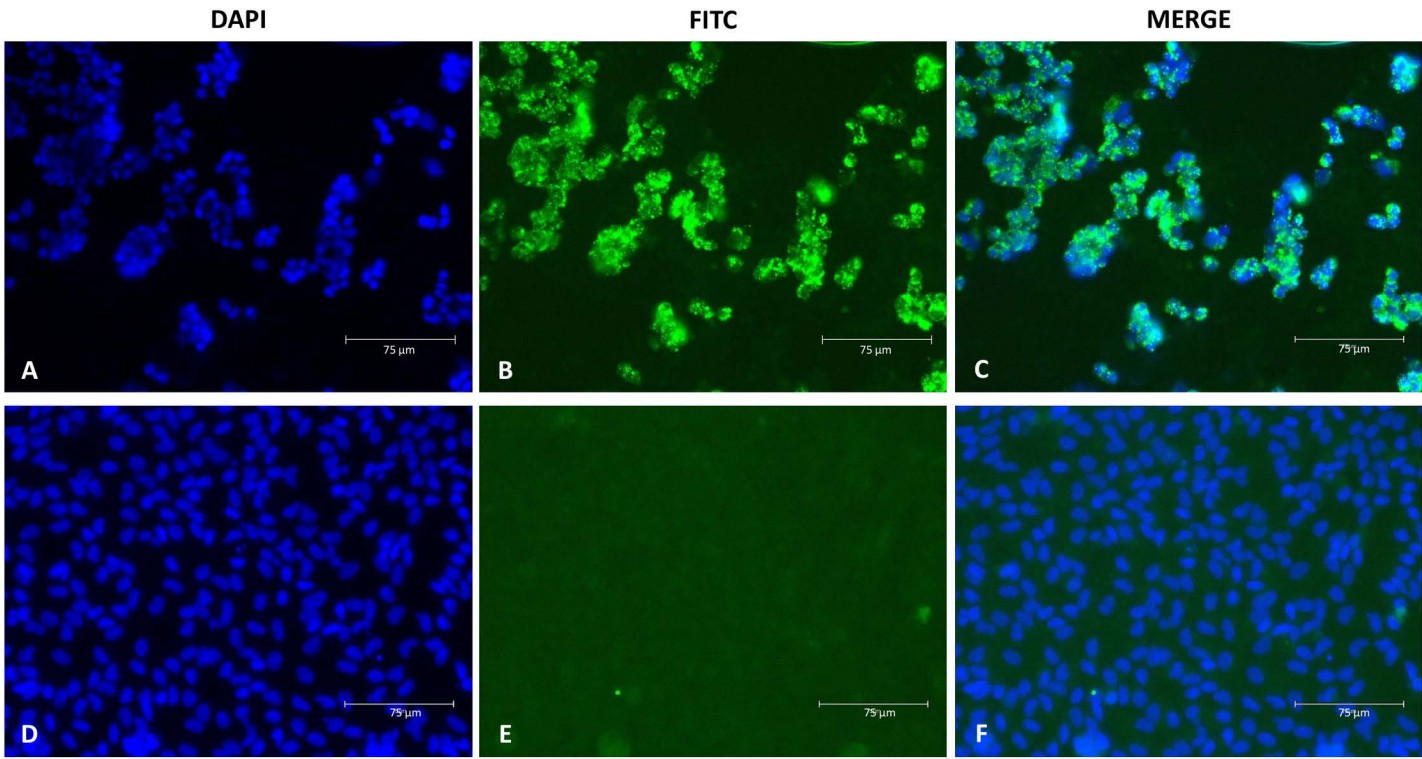

DAPI FITC MERGE

**Fig 1. Visualization of ARG-Epuyén by Indirect immunofluorescence assay.** The micrograph shows Vero E6 cells infected with the ANDV ARG-Epuyén strain (p4, 7 days post-infection), micrograph with DAPI, FITC and merge (A, B, C respectively). Vero E6 cell control (mock infected cells), equal condition (D, E, F respectively). Cells were stained with rabbit polyclonal serum against the nucleoprotein of ANDV, followed by an FITC-conjugated antibody against rabbit IgG. DAPI was used for DNA staining (4',6-diamidino-2-phenylindole). The images were captured at 400X magnification.

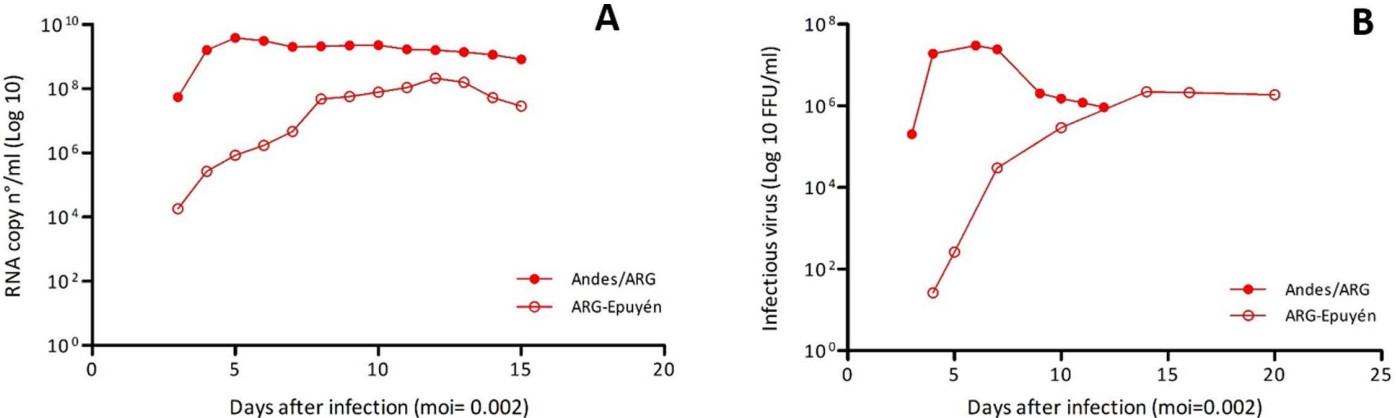

**Fig 2. Comparative kinetics curves of Andes/ARG and ARG-Epuyén strains in Vero E6.** Comparative levels of viral RNA (A) and viral particles (B) in culture supernatants of Vero E6 infected with Andes/ARG and ARG-Epuyén strains (MOI = 0.002).

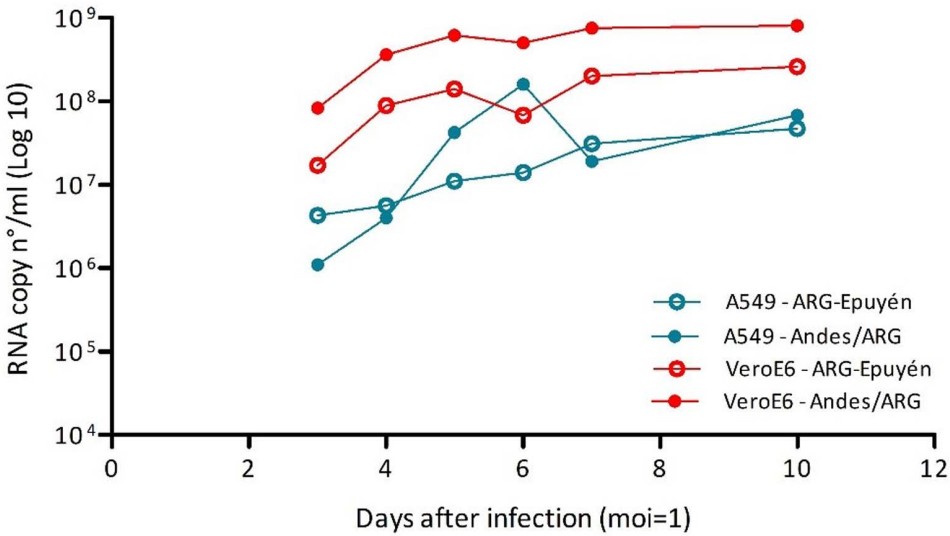

**Fig 3. Comparative kinetics curves of ANDV strains in Vero E6 and A549 cells.** RNA load was expressed as the number of copies of the S-segment/ml of supernatant. The curves show the comparative kinetics between the strains Andes/ARG and ARG-Epuyén in both Vero E6 and A549 (MOI = 1).

## Both strains of Andes virus are highly transmissible in Hamsters

Seven Syrian Golden hamsters of different sexes (3 males: MI; 4 females: FI) were inoculated intramuscularly with high doses of ANDV (approximately $10^4$ FFU) and placed in separate cages with one or two non-inoculated cage-mates (receptors: MR and FR). During the 30 days post-infection (p.i.) period, neither the hamsters inoculated with ARG-Epuyén (n = 6) nor the receptors exposed to them (n = 9) showed any visible signs of illness. However, infection was confirmed in all of them through seroconversion and/or genomic RNA detection in lung tissues (Table 1). In contrast, the single hamster inoculated with Andes/ARG (MI-1) exhibited inappetence and lethargy starting from day five p.i., and dyspnea was evident from day 13 to day 16 p.i. After 11 days of illness, MI-1 gradually improved and fully recovered by the end of the experiment. However, the two hamsters exposed to MI-1, MR-1 and MR-2, became ill 16 and 22 days after the onset of symptoms in MI-1, respectively, and both rapidly died. After 30 days p.i., all the survivors were sampled and sacrificed. The six index hamsters showed a strong humoral response (IgG titers >25,600), while four out of 11 of their cage-mates (receptors) exhibited low to moderate IgG titers. Nevertheless, all of them (n = 11) were infected, as evidenced by the detection of viral RNA in blood (up to $1.5 \times 10^7$ RNA copies/ml) and/or in lung tissues (up to $1.6 \times 10^8$ copies/100 ng RNA).

## The isolated ARG-Epuyén strain accumulated only two mutations

To analyze potential changes that could lead to viral adaptation for growth in cell culture and replication in hamsters, the complete genomes of the isolated virus (p4) and from the lung of an infected hamster (MI-3) were sequenced and compared with the previously published sequence from the human sample (GenBank acc. n° MN258238, MN258204, MN258171). The isolated strain ARG-Epuyén p4 showed only one nucleotide change in the non-coding region of the S segment (position 1522) and one non-synonymous change, K2116E, in the L segment (A6381G). No changes were observed between the isolated strain (p4) and the virus recovered from the infected hamster.

To investigate the molecular basis for the observed differences in pathogenicity between ARG-Epuyén (p4) and Andes/ARG (p19) in the hamster model, we compared the complete sequences of both strains. We identified 25

**Table 1. Evaluation of virulence of Andes virus strains in the golden hamster model. The necropsy and RNA extraction was realized after 30 days post infection, except for MR-1 and MR-2 which were realized on day 16 and 22 respectively. Enzyme-Linked Immunosorbent Assay (ELISA) for IgG titer was performed after the necropsy.**

| Cage N° | Infected | Receptor | Sex | ANDV strain | IgG titer | viral RNA in blood (copies/ml) | viral RNA in lung (copies/100 ng RNA) |
|---|---|---|---|---|---|---|---|
| 1 | | | | | | | |
| | MI-1 | | M | Andes/ARG | >25,600 | $7.4 \times 10^6$ | $1.35 \times 10^8$ |
| | | MR-1 | M | | <100 | NA | $8.5 \times 10^7$ |
| | | MR-2 | M | | <100 | NA | $5.2 \times 10^6$ |
| 2 | | | | | | | |
| | FI-2 | | F | ARG-Epuyén | >25,600 | $1.5 \times 10^7$ | $5.3 \times 10^3$ |
| | | FR-3 | F | | 400 | ND | $7.2 \times 10^6$ |
| 3 | | | | | | | |
| | MI-3 | | M | ARG-Epuyén | >25,600 | $1.5 \times 10^7$ | $1.5 \times 10^7$ |
| | | FR-4 | F | | 6400 | $7 \times 10^6$ | $1.6 \times 10^8$ |
| | | FR-5 | F | | <100 | $7.4 \times 10^6$ | $4.3 \times 10^7$ |
| 4 | | | | | | | |
| | FI-4 | | F | ARG-Epuyén | >25,600 | $4.1 \times 10^6$ | $1 \times 10^5$ |
| | | FR-6 | F | | <100 | ND | $5.8 \times 10^3$ |
| | | FR-7 | F | | 1600 | $1 \times 10^7$ | $7.8 \times 10^7$ |
| 5 | | | | | | | |
| | FI-5 | | F | ARG-Epuyén | >25,600 | $3.3 \times 10^6$ | $3.4 \times 10^7$ |
| | | FR-8 | F | | <100 | $1.8 \times 10^4$ | $3.9 \times 10^5$ |
| 6 | | | | | | | |
| | MI-6 | | M | ARG-Epuyén | >25,600 | $6.5 \times 10^6$ | $2.4 \times 10^7$ |
| | | MR-9 | M | | <100 | $2.9 \times 10^4$ | $5.5 \times 10^3$ |
| 7 | FI-7 | | F | ARG-Epuyén | >25600 | $4.1 \times 10^6$ | $8.7 \times 10^2$ |
| | | FR-10 | F | | 800 | $1.2 \times 10^5$ | $6.2 \times 10^6$ |
| | | FR-11 | F | | <100 | ND | $3 \times 10^4$ |
| 8 | | | | | | | |
| | MC-8 | | M | Mock-infected | 0 | 0 | 0 |
| | | MR-12 | M | | 0 | 0 | 0 |
| 9 | | | | | | | |
| | FC-9 | | F | Mock-infected | 0 | 0 | 0 |
| | | FR-13 | F | | 0 | 0 | 0 |

At day 30 post infection blood samples were collected from all the hamsters and then were sacrificed and necropsied. The exceptions were MR-1 and MR-2 which were found dead at day 16 and 22 respectively, then lung samples were obtained post mortem.

non-synonymous substitutions: three in the S segment (one in the N ORF and two in the NSs ORF), 10 in the M segment (six in Gn and four in Gc), and 12 in the L segment (Table 2).

## Discussion

The limited progress in vaccine development and other medical countermeasures (MCMs) against hantaviruses has been partly due to restricted access to wild-type viral strains and the lack of appropriate animal models of the disease. Additionally, no reverse genetic system for hantaviruses has been reported to date. Generally, *in vivo* and *in vitro* studies of hantavirus infections have been conducted using cell-cultured strains derived from moderate to high passage numbers. The success of MCM development depends on selecting strains that are actively circulating and have proven pathogenicity in humans.

**Table 2.  Evaluation of amino acid changes of Andes virus strains.**

| Genomic segment gene | Position | ANDV strain | | | | |
|---|---|---|---|---|---|---|
| | | CHI-9717869 | CHI-7913 | Andes/ARG, passage N°19 | Epuyén/18–19 (HPS case) | ARG-Epuyén, passage N°4 |
| **S-Segment** | | | | | | |
| GenBank acc. n° | | MT956622 | MT956618 | OP555722 | MN258238 | PQ215668 |
| | | | | | | |
| N | 21 | A | A | T | A | A |
| | | | | | | |
| NSs (ORF + 1) | 40 | Q | Q | Q | R | R |
| | 47 | N | N | N | S | S |
| **M-Segment** | | | | | | |
| GenBank acc. n° | | MT956623 | MT956619 | OP555727 | MN258204 | PQ215669 |
| Gn | 97 | S | S | P | S | S |
| | 114 | I | I | V | I | I |
| | 216 | F | F | F | L | L |
| | 353 | T | V | V | I | I |
| | 499 | V | V | V | I | I |
| | 641 | T | T | T | I | I |
| | | | | | | |
| Gc | 938 | T | A | T | A | A |
| | 1055 | S | S | A | S | S |
| | 1115 | V | V | V | I | I |
| | 1127 | V | V | I | V | V |
| **L-Segment** | | | | | | |
| GenBank acc. n° | | MT956621 | MT956620 | OP555734 | MN258171 | PQ215670 |
| RdRp | 141 | T | I | I | V | V |
| | 144 | R | R | K | R | R |
| | 364 | N | N | N | S | S |
| | 402 | I | V | V | I | I |
| | 541 | A | A | V | A | A |
| | 876 | S | S | S | A | A |
| | 1295 | I | I | M | I | I |
| | 1440 | S | S | S | N | N |
| | 1665 | V | I | I | V | V |
| | 1675 | P | P | S | P | P |
| | 1965 | Q | Q | Q | H | H |
| | 2113 | T | A | A | T | T |
| | 2116 | K | K | K | K | E |

Following the COVID-19 pandemic, the National Institute of Allergy and Infectious Diseases proposed a prototype pathogen approach to develop a generalizable MCM strategy. This approach can be applied to other viruses within the same viral family, enabling the rapid development of MCMs and shortening the timeline between pathogen outbreak and regulatory authorization if a virus with similar properties emerges. ANDV has been mentioned as a prototype pathogen of the *Hantaviridae* family and should be considered for vaccine development and pre-clinical and clinical testing [24,25]. For these purposes, the availability of well-characterized ANDV strains is critical.

In this work, we described the isolation of the strain responsible for the largest ANDV PTP transmission outbreak, which occurred in the small town of Epuyén and began on November 2, 2018. This strain, ARG-Epuyén, exhibited a high capacity for PTP transmission, necessitating the implementation of quarantine measures to curtail further spread [8]. The median reproductive number (the mean number of secondary cases caused by an infected person) was 2.12 before control measures were implemented and subsequently dropped to below 1.0 by late January. Early intervention allowed for the collection of samples leading to the isolation of this new ANDV strain from an asymptomatic case. An early passage of this strain was sequenced, revealing only one amino acid difference from the virus recovered from the patient. Like the Andes/ARG strain, this strain was able to grow in a new host without needing adaptation [26].

Two of the three previously isolated ANDV strains in cell culture caused lethal disease in hamsters [17–19,26]. Although this animal model has become crucial for studying the pathogenesis of HPS, the spreading capacity and horizontal transmission of ANDV have only been evaluated in one previous study with the CH-9717869 strain [27]. In this study, we assessed the pathogenicity and spreading capacity of the strain responsible for sustained PTP transmission in this animal model and compared it with the highly lethal Andes/ARG strain in direct exposure experiments. All hamsters—both those directly inoculated and those exposed to infected hamsters—became infected. Our data indicated that ARG-Epuyén was less virulent than Andes/ARG, as none of the six index hamsters inoculated with ARG-Epuyén developed severe disease during the 30-day period, despite showing high viral titres in blood and lungs. However, as five hamsters exposed to the index hamsters were sacrificed during the incubation period—when they had no IgG titres and rising viral RNA loads in blood and lungs—their final outcomes could not be assessed. We demonstrated that both Andes/ARG and ARG-Epuyén were highly transmissible in direct exposure experiments. Although the mechanism of transmission remains to be confirmed through indirect exposure experiments, we hypothesize that the route was likely respiratory or digestive, due to the absence of wounds in the animals. In a previous work, we identified in ARG-Epuyén three positions that are present in other two strains involved in PTP, R40 and S47 in the NSs, and I641 in the Gn [26]. However, to confirm if these positions confers the unusual capacity of spreading, it will be necessary to obtain more completes sequences from PTP cases.

Our preliminary data revealed differences in lethality between the two Argentinean strains. A previous comparative study of two Chilean strains found similar results. The strain obtained from a human serum sample (CHI-7913) caused an asymptomatic infection in hamsters, while the strain obtained from a rodent (CHI-9717869) was highly lethal [19]. The molecular basis for the observed differences in pathogenesis between these strains was evaluated through in silico studies [19,26]. However, the specific differences observed need further investigation through directed mutagenesis experiments. The N protein plays no role, as both Chilean strains share 100% identity. *In vitro* studies suggested that ANDV virulence could be determined by its ability to alter cellular signalling pathways by restricting the early induction of beta interferon (IFNB) and IFN-stimulated genes (ISGs) [26–28]. Additionally, compared with other pathogenic hantaviruses, ANDV is unique in that three viral proteins—N, NSs, and Gn/Gc—can block the same signalling pathway at different levels [29]. However, the few differences found in the Gn/Gc between the strains seem to be located outside conserved motifs or domains involved in the regulation of the IFNB response [30]. Further characterization of the NSs and RdRp proteins is needed to evaluate the effects of the observed changes.

As a result of comparing the complete sequences of both Argentinean ANDV strains, several amino acid changes were predicted. None of the substitutions represented drastic changes that could affect key motifs already described for the structural conformation and/or function of the N, Gn/Gc, and RdRp proteins. Exceptions included the changes P97S and T641I in the M-segment; the P residue is in the Gn ectodomain and may impact the flexibility of a loop structure, affecting recognition by some immune response components and/or interaction with cell receptors. The T641I change, located in an alpha-helix structure near the WAASA motif, may result in the loss of an oxygen molecule capable of forming a hydrogen bond with the adjacent residue C642.

In conclusion, we obtained a low-passaged isolate of a prototype pathogen: ANDV. The new strain, ARG-Epuyén, is a well-characterized isolate due to its known outcome in a recent HPS outbreak and its high PTP transmission potential.

Furthermore, the isolation was obtained directly from a clinical sample with a low number of passages (p = 4) and only one amino acid change. Future studies will be necessary to determine if these two changes are attenuation determinants for the hamster model, or whether they lack pathogenic impact and therefore this strain could be considered an ANDV wild-type strain. If so, it would represent a critical tool toward developing MCMs against this emerging pathogen.

## Supporting information

**S1 Table. Comparative kinetics curves of Andes/ARG and ARG-Epuyén strains in Vero E6.** Comparative levels of viral RNA (S1A) and viral particles (S1B) in culture supernatants of Vero E6 infected with Andes/ARG and ARG-Epuyén strains (MOI = 0.002).
(XLSX)

**S2 Table. Comparative kinetics curves of ANDV strains in Vero E6 and A549 cells.**
(XLSX)

## Acknowledgments

We thank Silvia Girard and Lara Martin for laboratory support.

## Author contributions

**Conceptualization:** Valeria Paula Martinez.

**Formal analysis:** Rocio Coelho, Sebastian Kehl, Daniel Alonso, Carla Bellomo, Valeria Paula Martinez.

**Investigation:** Rocio Coelho, Sebastian Kehl, Natalia Periolo, Daniel Alonso, Darío Fernández Do Porto, Alexis Edelstein, Carla Bellomo, Valeria Paula Martinez.

**Methodology:** Rocio Coelho, Sebastian Kehl, Natalia Periolo, Daniel Alonso, Carla Bellomo, Valeria Paula Martinez.

**Resources:** Emiliano Biondo, Celeste Perez, Darío Fernández Do Porto, Gustavo Palacios, Alexis Edelstein.

**Supervision:** Carla Bellomo, Valeria Paula Martinez.

**Visualization:** Natalia Periolo.

**Writing – original draft:** Rocio Coelho, Sebastian Kehl, Valeria Paula Martinez.

**Writing – review & editing:** Valeria Paula Martinez.

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
