## [Decision Letter · Decision Letter 0]

Thank you for submitting your manuscript to PLOS Neglected Tropical Diseases. After careful consideration, we feel that it has merit but does not fully meet PLOS Neglected Tropical Diseases's publication criteria as it currently stands. Therefore, we invite you to submit a revised version of the manuscript that addresses the points raised during the review process.

Please submit your revised manuscript within 30 days Jun 08 2025 11:59PM. If you will need more time than this to complete your revisions, please reply to this message or contact the journal office at plosntds@plos.org. Response to Reviewers Revised Manuscript with Track Changes Manuscript

Shaden Kamhawi

co-Editor-in-Chief

Paul Brindley

co-Editor-in-Chief

**Journal Requirements:**

- ® on page: 7.

3) Thank you for including an Ethics Statement for your study. Please include:

i) The full name(s) of the Institutional Review Board(s) or Ethics Committee(s)

ii) The approval number(s), or a statement that approval was granted by the named board(s).

4) Tables should not be uploaded as individual files. Please remove these files from the online submission form. For more information about how to format tables, see our guidelines:

https://journals.plos.org/plosntds/s/tables 

5) We note that your Data Availability Statement is currently as follows: "All relevant data are within the manuscript and its Supporting Information files". There are not any files uploaded as supporting information. Please confirm at this time whether or not your submission contains all raw data required to replicate the results of your study. Authors must share the “minimal data set” for their submission. PLOS defines the minimal data set to consist of the data required to replicate all study findings reported in the article, as well as related metadata and methods (https://journals.plos.org/plosone/s/data-availability#loc-minimal-data-set-definition).

6) The file inventory includes files for Figures 2a, and 2b. We would recommend either combining these into a single Figure 2.tiff file with separate internal panels, or renumbering them as individual figures, as we are not able to publish multiple components of a single figure as separate files.

**Reviewers' comments**

**Key Review Criteria Required for Acceptance?**

**Methods**

-Are the objectives of the study clearly articulated with a clear testable hypothesis stated?

-Is the study design appropriate to address the stated objectives?

-Is the population clearly described and appropriate for the hypothesis being tested?

-Is the sample size sufficient to ensure adequate power to address the hypothesis being tested?

-Were correct statistical analysis used to support conclusions?

-Are there concerns about ethical or regulatory requirements being met?

Reviewer #1: 1. Reference 22 is very important, as it is listed as the reference for the sequencing approach, but the reference is incorrect—it is a bioRxiv but I believe is now published in PLOS NTD.

2. Line 249, what MOI was used?

3. Line 125-142, what size flasks were used for infection at each step?

4. Where was the anti-ANDV serum obtained and how was it made?

Reviewer #2: The objectives and design of the study are well-defined.

Regarding concerns about regulatory aspects, the manuscript states that the experiments in the animal model were conducted in a safety level 3 facility; BSL4 is typically required.

Reviewer #3: The objective of the study is clear and well design to address the Virological characterization of new Andes virus strain and to support the conclusions. All experiments are presented, not in detail, but well referenced. There are no concerns about ethical requirements, but I ask for the number of the protocals and permissions to be included in the text.

**Results**

-Does the analysis presented match the analysis plan?

-Are the results clearly and completely presented?

-Are the figures (Tables, Images) of sufficient quality for clarity?

Reviewer #1: 1. Figure 1 legend states there are panels A-F, but there is only A and B on the figure itself.

2. Figure 2, why are there no time points for ARG after day 12 like Epuyen?

3. There is no text in the Figure 2 and Figure 3 figure legends.

4. FFU data is needed for Figure 3.

5. Table 1 should describe at what timepoints post-infection the viral RNA was tested in lung and blood and when the IgG titers were tested.

6. Please show negative controls for ANDV staining in Figure 1.

Reviewer #2: The analysis and results are written in an orderly and clear manner, with their respective subheadings.

The tables and figures are highly relevant, well-prepared, and easy to understand.

Table 1 establishes that there are 18 rodents in cages 1 to 7, and the text establishes that there are 17 in 7 cages.

Reviewer #3: All results match the analysis plan, and are clearly presented. With the exception of figure 1 all images and tables are of sufficient quality.

**Conclusions**

-Are the conclusions supported by the data presented?

-Are the limitations of analysis clearly described?

-Do the authors discuss how these data can be helpful to advance our understanding of the topic under study?

-Is public health relevance addressed?

Reviewer #1: 1. Lines 324-326, 377, and 282, can it be stated that the isolated virus is a wild-type strain and one that doesn’t need adaptation for in vitro growth? It has a mutation in the non-coding region of S and a coding mutation in L. As a comparison, the vaccine TC-83 strain of Venezuelan equine encephalitis virus is attenuated due to only 2 mutations compared to the wild-type strain—one in the 5’ UTR and an amino acid change in the glycoprotein. While the number of changes is small in this manuscript after isolation—and impressively so, given the difficulty of isolating hantaviruses—some nuance should be used to discuss “wild-type” and “adaptation.”

Reviewer #2: I believe the discussion is appropriate and well-informed; I would suggest focusing the discussion more on transmission (person-to-person) than on pathogenicity.

It would be interesting to know the authors' opinion on the observation that a significant viral load is detected in some rodents, yet no antibody response is observed.

Reviewer #3: The conclusions are supported by the data, limitations are included in the discussion section.

Throughout the text the authors emphasized the how these data can be helpful to advance and enhance medical

countermeasures with a clear understand of how basic science can impact public health.

**Editorial and Data Presentation Modifications?**

Reviewer #1: 1. The manuscript should be reviewed for English grammar and style. Below are a few changes that should be made:

Line 77, please change “After infection, the signs and symptoms of the disease can manifest after a long period of up to 40 days” to “After human infection, the signs and symptoms of the disease can manifest after a long incubation period of up to 40 days”

Line 89, change “could be” to “was”

Line 99, change “since” to “and”

Reviewer #2: (No Response)

Reviewer #3: Line 21 Andes virus (Orthohantavirus andesense)

Line 47 but none of them were

Line 49 and has an unclear history

Line 58 please refer to the updated taxonomy of hantaviruses (Bunyaviricetes: Elliovirales: Hantaviridae: Mammantavirinae)

Line 112 use PTP that has already been cited in the text

Line 117 use HPS that has already been cited in the text

Line 165 how many days? Please specify

Line 171 … transmission, Syrian Golden Hamsters (Mesocricetus auratus)

Line 197 TRIzol

Line 218/19 – provide the protocols number for the ethics permissions

Line 223 - how many days after the probable exposure?

Table 2 – please include GenBank accession number

Line 299/300 – use MCMs that have already been cited in the text

Line 312 – Hantaviridae needs to be in italic

Line 318 2nd

Throughout the text the terms “in vitro” and “in vivo” need to be in italic

**Summary and General Comments**

Reviewer #1: The manuscript by Coelho and Kehl et al. describes the isolation and characterization in a hamster model of a strain of Andes hantavirus that caused the largest person-to-person Andes virus outbreak to date. There is thorough detail in the isolation of the virus, a difficult procedure which will be very useful for future hantavirus studies. There is also description of the in vivo infectivity of this virus in hamsters, which appears to be non-pathogenic but capable of animal-to-animal transfer. The manuscript would be improved with some changes as suggested in this review.

Reviewer #2: Isolating the Andes hantavirus from a human is a significant achievement, even more so if this isolation comes from a patient involved in a person-to-person outbreak of the magnitude of the one that occurred in southern Argentina. It is no exaggeration to say that this isolation will contribute to a better understanding of the epidemiology of the infection and the development of therapeutic and control tools. The paper is well-written, with well-founded statements, and the study design and methodology are appropriate. The characterization of the infection dynamics in cell lines, the genetic studies with sequencing, and the evaluation of virulence and transmissibility in the Syrian hamster animal model were very well executed and provide relevant information.

The only general criticism that can be raised about this work is that the title may suggest that the objective of the study is to try to elucidate whether the virus involved in a person-to-person outbreak has some special characteristic that explains why this virus acquires a capacity that is unusual in hantaviruses. I would suggest that the discussion analyze whether some of the substitution changes in the M segment could have made the virus capable of transmitting to humans, or speculate on what circumstances or virological properties could have led to this epidemiological event.

It is very interesting that the viruses tested are transmissible between hamsters, and intriguing that the virus isolated from a human does not cause disease in this model. Both results (transmissibility and lack of disease development with human viruses) confirm previous findings, adequately described in the paper.

Reviewer #3: The work presented here by Coelho et al. brings a very interesting view on person to person transmission of hantavirus, a fallow up on the 2018 outbreak focused on virological characterization and understand of a new Andes virus strain ability to be shared between humans. The manuscript is well designed and have sound methodology using classical and modern assays to accomplish their objectives. The results raise a series of questions in how person to person is spread, but that is not the goal of this manuscript. This is a much need study so we can be a step closer to understand Andes virus unique features. In opinion the article is ready for publication with only some minor revisions and actualization listed bellow:

Line 21 Andes virus (Orthohantavirus andesense)

Line 47 but none of them were

Line 49 and has an unclear history

Line 58 please refer to the updated taxonomy of hantaviruses (Bunyaviricetes: Elliovirales: Hantaviridae: Mammantavirinae)

Line 112 use PTP that has already been cited in the text

Line 117 use HPS that has already been cited in the text

Line 165 how many days? Please specify

Line 171 … transmission, Syrian Golden Hamsters (Mesocricetus auratus)

Line 197 TRIzol

Line 218/19 – provide the protocols number for the ethics permissions

Line 223 - how many days after the probable exposure?

Table 2 – please include GenBank accession number

Line 299/300 – use MCMs that have already been cited in the text

Line 312 – Hantaviridae needs to be in italic

Line 318 2nd

Throughout the text the terms “in vitro” and “in vivo” need to be in italic

PLOS authors have the option to publish the peer review history of their article (what does this mean? ). If published, this will include your full peer review and any attached files.

**Do you want your identity to be public for this peer review?** For information about this choice, including consent withdrawal, please see our Privacy Policy .

Reviewer #1: No

Reviewer #2: **Yes: ** Pablo Agustín Vial

Reviewer #3: No

**Figure resubmission:****Reproducibility:** To enhance the reproducibility of your results, we recommend that authors of applicable studies deposit laboratory protocols in protocols.io, where a protocol can be assigned its own identifier (DOI) such that it can be cited independently in the future. Additionally, PLOS ONE offers an option to publish peer-reviewed clinical study protocols. Read more information on sharing protocols at https://plos.org/protocols?utm_medium=editorial-email&utm_source=authorletters&utm_campaign=protocols

---

## [Decision Letter · Decision Letter 1]

Dear Bsc Biological Sciences Kehl,

We are pleased to inform you that your manuscript 'Virological characterization of a new isolated strain of Andes virus involved in the recent person-to-person transmission outbreak reported in Argentina' has been provisionally accepted for publication in PLOS Neglected Tropical Diseases.

Best regards,

Jonas Klingström

Academic Editor

Andrea Marzi

Section Editor

Shaden Kamhawi

co-Editor-in-Chief

Paul Brindley

co-Editor-in-Chief

Reviewer's Responses to Questions

**Key Review Criteria Required for Acceptance?**

**Methods**

-Are the objectives of the study clearly articulated with a clear testable hypothesis stated?

-Is the study design appropriate to address the stated objectives?

-Is the population clearly described and appropriate for the hypothesis being tested?

-Is the sample size sufficient to ensure adequate power to address the hypothesis being tested?

-Were correct statistical analysis used to support conclusions?

-Are there concerns about ethical or regulatory requirements being met?

Reviewer #1: (No Response)

Reviewer #2: ok

**Results**

-Does the analysis presented match the analysis plan?

-Are the results clearly and completely presented?

-Are the figures (Tables, Images) of sufficient quality for clarity?

Reviewer #1: (No Response)

Reviewer #2: ok

**Conclusions**

-Are the conclusions supported by the data presented?

-Are the limitations of analysis clearly described?

-Do the authors discuss how these data can be helpful to advance our understanding of the topic under study?

-Is public health relevance addressed?

Reviewer #1: (No Response)

Reviewer #2: ok

**Editorial and Data Presentation Modifications?**

Reviewer #1: (No Response)

Reviewer #2: ok

**Summary and General Comments**

Reviewer #1: The authors did a nice job responding to my critiques, and I feel the manuscript is acceptable for publication.

Reviewer #2: ok

PLOS authors have the option to publish the peer review history of their article (what does this mean? ). If published, this will include your full peer review and any attached files.

**Do you want your identity to be public for this peer review?** For information about this choice, including consent withdrawal, please see our Privacy Policy .

Reviewer #1: No

Reviewer #2: **Yes: ** Pablo Vial

---

## [Editor Report · Acceptance letter]

Dear Bsc Biological Sciences Kehl,

We are delighted to inform you that your manuscript, "Virological characterization of a new isolated strain of Andes virus involved in the recent person-to-person transmission outbreak reported in Argentina," has been formally accepted for publication in PLOS Neglected Tropical Diseases.

Best regards,

Shaden Kamhawi

co-Editor-in-Chief

Paul Brindley

co-Editor-in-Chief
